Regularizing priors for Bayesian VAR applications to large ecological datasets

http://orcid.org/0000-0002-4359-0296 Ward Eric J. 1 eric.ward@noaa.gov
Marshall Kristin 2
Scheuerell Mark D. 3
1 Conservation Biology Division, Northwest Fisheries Science Center, National Marine Fisheries Service, NOAA , Seattle, WA , United States
2 Fishery Resource Analysis and Monitoring Division, Northwest Fisheries Science Center , Seattle, WA , USA
3 U.S. Geological Survey Washington Cooperative Fish and Wildlife Research Unit, School of Aquatic and Fishery Sciences, University of Washington , Seattle, WA , USA
Letcher Benjamin
Electronic publication date: 2022 Nov 8
Publication date: 2022
Volume: 10
Electronic Location ID: e14332
Received 2022 Jul 26; Accepted 2022 Oct 11
Copyright: © 2022 Ward et al.
Copyright year: 2022
Copyright holder: Ward et al.
License: This is an open access article distributed under the terms of the Creative Commons Attribution License, which permits unrestricted use, distribution, reproduction and adaptation in any medium and for any purpose provided that it is properly attributed. For attribution, the original author(s), title, publication source (PeerJ) and either DOI or URL of the article must be cited.
License URL: https://creativecommons.org/licenses/by/4.0/

Keywords: Bayesian lasso, Spike-slab, Regularization, Shrinkage, VAR, VARSS, Community dynamics, Multivariate regression, Big data, Variable selection

Funding: Northwest Fisheries Science Center Kristin Marshall was supported on a National Research Council (NRC) post-doctoral fellowship at the Northwest Fisheries Science Center while this research was performed. The funders had no role in study design, data collection and analysis, decision to publish, or preparation of the manuscript.

==============================
Using multi-species time series data has long been of interest for estimating inter-specific interactions with vector autoregressive models (VAR) and state space VAR models (VARSS); these methods are also described in the ecological literature as multivariate autoregressive models (MAR, MARSS). To date, most studies have used these approaches on relatively small food webs where the total number of interactions to be estimated is relatively small. However, as the number of species or functional groups increases, the length of the time series must also increase to provide enough degrees of freedom with which to estimate the pairwise interactions. To address this issue, we use Bayesian methods to explore the potential benefits of using regularized priors, such as Laplace and regularized horseshoe, on estimating interspecific interactions with VAR and VARSS models. We first perform a large-scale simulation study, examining the performance of alternative priors across various levels of observation error. Results from these simulations show that for sparse matrices, the regularized horseshoe prior minimizes the bias and variance across all inter-specific interactions. We then apply the Bayesian VAR model with regularized priors to a output from a large marine food web model (37 species) from the west coast of the USA. Results from this analysis indicate that regularization improves predictive performance of the VAR model, while still identifying important inter-specific interactions.

Introduction

Across a wide range of statistical tools—ranging from simple linear regression to complicated spatiotemporal models—a fundamental question in ecology, fisheries, and related fields is identifying a subset of important predictor variables from a larger set of potential explanatory variables. These types of statistical analyses are often constrained by the “small n, large p” problem (West, 2003). For example, in basic linear regression analyses, the number of estimated parameters p cannot exceed the sample size n, because the degrees of freedom (n − p) is constrained to be greater than 0 (Zar, 1999). Furthermore, as p approaches n, the ability to estimate parameter uncertainty also diminishes. Similar constraints exist for hierarchical or mixed effects models, but calculating degrees of freedom becomes more complex (Spiegelhalter et al., 2002; Bolker et al., 2009).

Like other fields, ecology has recently undergone a “big data” revolution (Howe et al., 2008; Hampton et al., 2013b). Movement towards managing entire ecosystems rather than single species has spurred large-scale monitoring efforts and efforts to synthesize multiple associated data streams (Harvey et al., 2018). Simultaneously, greater ecosystem complexity has been incorporated in simulation models used for natural resource management (Sitch et al., 2003; Fulton, Smith & Johnson, 2003; Crowder & Norse, 2008). Regardless of whether inference is being made from observational data or simulation results, statistical models fit to these data may be challenged by the sample size. A classic example of a family of ecological models that has been limited by large streams of data are vector autoregressive (VAR) models (Hampton et al., 2013a). Ecologists use these models to estimate species interactions from observed multivariate time series (Ives et al., 2003; Holmes, Ward & Wills, 2012), and a general challenge of their use is that the number of pairwise interactions in a community grows proportionately to the square of the number of species (Ovaskainen et al., 2017). In addition to making inference about relationships between species, these interactions can also be used to derive metrics of stability and resilience (Ives et al., 2003).

A number of dimension reduction approaches have been used in ecology and related fields to reduce many potential predictor variables to a subset of variables with high explanatory and predictive power. Popular examples include stepwise regression (Hocking, 1976) or all-subsets regression (Miller, 2002), and both are widely available in several R packages (R Core Team, 2022); examples include ‘step’ in stats, ‘stepAIC’ in MASS (Venables & Ripley, 2002), ‘dredge’ in MuMIn (Bartoń, 2020), and ‘regsubsets’ in leaps (Miller, 2020). Both stepwise and all subsets regression have widely documented shortcomings, including violating assumptions about multiple hypothesis testing (Whittingham et al., 2006; Mundry & Nunn, 2009) and the potential to identify spurious correlations (Olden & Jackson, 2000; Anderson et al., 2001), but they continue to be widely used.

In statistics, machine learning, and related fields, penalized regression has been used as an alternative technique to reduce model complexity (Hoerl & Kennard, 1970; Tibshirani, 1996; O’Hara & Sillanpää, 2009). Penalized regression consists of finding the combination of parameters that minimizes the objective function g(θ)=∑i=1n⁡(Yi−Y^(θ)i)2+P, where Yi and Y^(θ)i are the ith observed and estimated data points, respectively; θ represents the regression coefficients; and P is a penalty term. For ordinary least squares regression, P=0, and g(θ)reduces to the traditional sum of squares. Many choices for P exist, and are similar in that the further regression coefficients deviate from 0, the greater the penalty. One form known as ridge regression applies a quadratic or ‘L2’ penalty, P=λ∑j=1m⁡θj2, where λ is a shrinkage parameter that controls the degree of regularization (Hoerl & Kennard, 1970). A second approach, known as lasso regression (least absolute shrinkage and selection operator), involves applying a ‘L1’ penalty of P=λ∑j=1m⁡|θj|. For both ridge and lasso methods, as λ increases in magnitude, the penalty for the regression coefficients departing from zero also increases (Tibshirani, 1996). With many sparse coefficients, the advantage of using lasso regression is that absolute penalties of small values are greater than quadratic penalties, implemented in ridge regression (Wu & Lange, 2008). Thus, while lasso regression penalizes coefficients to zero, ridge regression doesn’t penalize coefficients to exactly zero.

By placing a greater penalty on model complexity compared to standard ordinary least squares (OLS) regression, a subset of estimated coefficients in penalized regression become fixed at 0. This yields models that have better predictive accuracy than OLS estimates (Tibshirani, 1996). A challenge in implementing penalized regression techniques is that the regularization parameter λ needs to be chosen or estimated. Routines for comparing values of λ can be compared via cross-validation with bootstrapped datasets. Like stepwise or all subsets regression, these methods are available in several R packages; examples include ‘lars’ to implement least angle regression (Efron et al., 2004), ‘elasticnet’ to implement a hybrid L1/L2 penalization (Zou & Hastie, 2005), ‘penalized’ (Goeman, Meijer & Chaturvedi, 2018), and ‘glmnet’ (Friedman, Hastie & Tibshirani, 2010). Several applications of these methods exist in the context of VAR models (e.g., BigVAR, Nicholson, Matteson & Bien, 2019), though these have generally been developed in a maximum likelihood setting.

In addition to the maximum likelihood approaches, Bayesian lasso methods have been developed that treat the regularization parameter λ as an estimated hyper-parameter; by integrating over values of λ via Markov Chain Monte Carlo (MCMC), robust coefficient estimates that are marginalized over values of λ can be generated (Casella et al., 2010). Mechanistically, this involves specifying double-exponential or Laplace priors on regression coefficients (Park & Casella, 2008; O’Hara & Sillanpää, 2009). Alternative Bayesian priors to the lasso include mixture or “spike-slab” priors (Miller, 2002; reviewed by O’Hara & Sillanpää, 2009). Spike-slab priors on potentially sparse coefficients model the prior variance as a mixture of a wide distribution with high variance (the “slab”) and a narrow distribution with small variance (the “spike” near zero). The contribution of each component can either be fixed a priori or estimated; challenges in implementing this type of shrinkage prior is that data-specific tuning is often required to ensure mixing between the two distributions, and results may be sensitive to the choice of tuning parameters (O’Hara & Sillanpää, 2009). Because of computational challenges that may occur when using the spike-slab prior, continuous alternatives have been a focus of recent development. Continuous priors may include regularization by using hyperparameters on variance terms (e.g., normal or Student-t distributions), or more flexible choices such as the horseshoe prior (Carvalho, Polson & Scott, 2010). These priors allow for both wide tails and high density near zero; because of their flexibility and scalability (Piironen & Vehtari, 2017), these priors have been incorporated into a number of software packages and are becoming widespread.

The objectives of our paper are to extend regularizing priors to Bayesian VAR models for ecological applications and develop software to implement these methods. We explore a range of potential priors for off-diagonal coefficients; examples include a regularized horseshoe prior, as well as simpler priors with estimated hyperparameters (normal and Student-t distributions). The sensitivity of model estimates to the choice of prior is evaluated using simulated data for models with and without observation error. As a case study, we compare the performance of these Bayesian regularization techniques to a high dimensional VAR model explaining the dynamics of 37 marine species from the California Current in the North East Pacific Ocean. All code for these models is deployed as a publicly available ‘varlasso’ R package, https://github.com/atsa-es/varlasso (Ward, Marshall & Scheuerell, 2022).

Methods

Vector autoregressive state space models

Vector autoregressive (VAR) models have been widely used in fisheries and related fields (these approaches are also known as multivariate autoregressive or MAR models). In the ecological literature, these are also referred to as the discrete time multivariate Gompertz models (Mutshinda, O’Hara & Woiwod, 2009). The VAR model consists of a process equation, xt+1=Bxt+u+wt, where xt is an m × 1 vector of log-abundances for species at time t, u is an m × 1 vector of species-specific growth rates or trends, B represents a m × m matrix of community interactions (element Bi,j describes the per-capita effect of species j on species i), and wt represents an m × 1 vector of random environmental effects at time t (Ives et al., 2003; Scheef et al., 2012). We assume environmental stochasticity is multivariate normal, such that wt∼MVN(0,Q), and Q may be a diagonal variance-covariance matrix (species have independent dynamics) or include correlation between species. The basic VAR model can be modified to also incorporate observation error model (yielding a state-space, or VARSS model). The observation equation relates the true states of nature at time t ( xt) to the observed data ( yt), yt=xt+vt, where vt∼MVN(0,R), and R represents the variance-covariance matrix of observation errors (Holmes, Ward & Wills, 2012). In addition to partitioning the total variance into process and observation errors, the VARSS model is flexible in that it is better suited for datasets with lots of missing values. In contrast, only abundance estimates that are adjacent in time contribute to the likelihood for the simpler VAR model (Ives et al., 2003).

Simulated data

We simulated datasets using estimated interactions from a simplified lake food web with four species groups described by (Ives et al., 2003). The interaction matrix for the low-planktivory system from Ives et al. (2003) is typical of many ecological applications in that (1) interspecific interactions (off-diagonal elements) are generally weaker than intraspecific interactions (density dependence, diagonal elements) and (2) a relatively large number of elements are 0 (8 of 16, Table S1).

We treated process errors as independent and identically distributed, such that Q=σpro2I, with σpro fixed at 0.2. For simulations focused on VAR models, observation error was not included. Observation error was added for VARSS simulations, with observation errors also assumed to be independent and identically distributed, such that, R=σobs2I. To explore several ratios of σobs:σproc, we varied σobs across three levels (0.05, 0.1, 0.2). For each combination of observation and process variance, we used 200 replicate datasets, each consisting of 40 timesteps. To ensure time series were approximately stationary, we performed a ‘burn-in’ of 200 timesteps for each, retaining the last 40 data points.

Priors

To compare the effects of regularizing priors, we applied three estimation models to each of our simulated datasets, varying only the prior formulations for the off-diagonal elements of the B matrix. Each estimation model assigned Bi,i∼Normal(0.7,1) priors to diagonal elements of B (representing intraspecific interactions) and truncated Normal(0.0,0.5) priors to the observation and process standard deviations ( σobs, σpro). We assumed that both process and observation errors were uncorrelated across taxa, so that Q=σpro2I and R=σobs2I.

Our three alternative formulations for priors on the off-diagonal elements of B were:

(1) Normal distribution

We implemented normal priors on off-diagonal elements to represent the status quo for Bayesian VAR models (Mutshinda et al., 2019). In this approach, Bi,j∼Normal(0.0,σN) and σN is assumed known. A slight deviation from Normal priors is to use Student-t priors, which can generate similar distributions to the Normal with large degrees of freedom ( ν), but also place more density on extreme values. In contrast to the unpooled approach where Bi,j are estimated independently, a partial pooling approach may be used with either the Normal or Student-t distribution to shrink estimates toward a common mean (in this case, 0). Partial pooling can be implemented by assigning a hyper-prior to σN. We include support for the Student-t distribution and partial pooling in our `varlasso` R package, but they are not included in our simulation analyses.

(2) Laplace distribution

As a second prior, we used a Laplace or double exponential prior (O’Hara & Sillanpää, 2009; Casella et al., 2010) (Fig. 1). Relative to the Normal distribution, the Laplace can place greater density near 0, and is controlled by a single parameter that controls the variance, Bi,j~Laplace(0.0,τ). An equivalent parameterization is as a mixture, α∼Exp(12τ2) where Bi,j∼Normal(0.0,α) (Ding & Blitzstein, 2018).

Figure 1 Illustration of three potential prior distributions for interactions in VAR and VARSS models.

All three priors are centered on 0 and their standard deviations are equal (0.24).

(3) Regularized horseshoe prior

As our third prior, we implement regularized horseshoe priors (Piironen & Vehtari, 2017) (Fig. 1). We use the same implementation as rstanarm and brms (Bürkner, 2017; Goodrich et al., 2020), so that priors on off-diagonal elements of B are Bi,j∼Normal(0.0,τ2λ~i,j2). The hyperparameter τ is assigned a τ∼Student−t(υ=3,0,ϕ) where ϕ is a global scale parameter and λ~i,j2 controls the regularization for the effect of species j on species i in the B matrix. The degree of regularization is allowed to be unique by modeling it as λ~i,j2=c2λi,j2c2+τ2λi,j2, and λi,j2 are treated as parameters with priors λi,j2∼Cauchy(0,1). The width of the slab (allowing for large B coefficients) is assigned a prior c2∼InvGamma(α=υslab2,β=sslab2υslab2), where υslab is the degrees of freedom and sslab2 is the scale of the slab.

Hyperparameters

We carefully selected hyperparameters for each of the three prior formulations, to ensure that the priors would have the same target standard deviation. Starting with the regularized horseshoe prior, we followed the advice of (Piironen & Vehtari, 2017), and we held the global df = 1. The same authors recommend quantifying ϕ (the global scale hyperparameter) as the ratio of non-zero coefficients to coefficients that are zero to the square root of the number of observations. Because of the multivariate nature of a VAR model, we constructed several preliminary scenarios letting the global scale range from 0.025 to 0.08 (focusing here with ϕ = 0.025). We also used these preliminary model runs to consider several combinations of horseshoe slab parameters; based on these simulations, we used υslab = 5 and sslab2 = 1.0. Combined, these choices of hyperparameters resulted in a prior with a standard deviation ~0.24; as a result we used values of υLP=3 and ϕLP=0.165 for the Laplace prior (allowing for wide tails but the same standard deviation), and fixed σN=0.24for the Normal prior.

Estimation

Estimation was done in a Bayesian framework using our varlasso R package (Ward, Marshall & Scheuerell, 2022). This package is built in R (R Core Team, 2022) and acts as an interface to Stan (Stan Development Team, 2022), which implements MCMC using the No-U Turn Sampling (NUTS) algorithm (Hoffman & Gelman, 2014; Carpenter et al., 2017). For all models, we ran three parallel MCMC chains, discarding the first 2,000 iterations of each and retaining the last 1,000 samples. In addition to using visual diagnostics (Gabry & Veen, 2018), we calculated effective sample size and R-hat statistics to evaluate convergence (Vehtari et al., 2021).

Quantifying performance

We used two metrics to quantify the performance of alternative model formulations, as there may be multiple objectives of VARSS analyses. We first calculated the Leave-One-Out Information Criterion (LOOIC) in the ‘loo’ R package (Vehtari, Gelman & Gabry, 2017; Vehtari et al., 2020). The LOOIC statistic represents an approximation to leave-one-out cross-validation, or overall predictive ability of the model. For any given dataset, LOOIC differences can be compared between models, with lower values corresponding to models with better predictive ability. To place LOOIC values on a relative scale across simulated datasets, we differenced LOOIC across priors for a given dataset relative to the lowest LOOIC value for that dataset. As a second metric, we computed log-score statistics to quantify the prior influence on B matrix parameter estimates. Log-scores are often used to quantify the bias and precision of predictions (Gneiting & Raftery, 2007); similarly, they can also be used to quantify the predictions of parameter estimates when true values are known, as in the case of our simulations. The log-score can be calculated a number of ways, but involves evaluating an observation (or parameter value) y across a predictive density f(y,θpost)=1n∑i=1:n⁡f(y|θi,post) where θpost is a vector containing samples from the posterior. If the density f() does not have a closed form, an alternative approach is to estimate the empirical CDF to approximate f() (Krüger et al., 2021). We adopted this empirical approach, using the ’scoringRules’ R package (Jordan, Krüger & Lerch, 2019) and calculated log-scores, log(f(θtrue,θpost)). Using this approach, higher log scores represent better agreement between posterior parameter estimates and truth.

Application to marine food webs

To demonstrate the utility of Bayesian regularization, we applied the approach described above to a VAR model describing the ecosystem dynamics of the California Current (Horne et al., 2010; Kaplan et al., 2013). The ‘Atlantis’ ecosystem modeling framework (Fulton et al., 2004) couples output from a hydrodynamic Regional Ocean Modeling System (ROMS) model with a spatially explicit food web model that may include 60+ functional groups, and includes fishing mortality. We used an Atlantis model implemented to represent the California Current marine ecosystem, including the fisheries it supports (Hermann et al., 2009; Horne et al., 2010; Kaplan et al., 2013; Marshall, Kaplan & Levin, 2014). Estimates of fish biomass for the California Current Atlantis Model are derived from fisheries stock assessments, survey indices, and published data on growth, life history, and food habits (Horne et al., 2010; Kaplan, Horne & Levin, 2012). We used the baseline model configuration from Marshall, Kaplan & Levin (2014) to generate ecosystem dynamics over a 50-year horizon. While this California Current Atlantis operates on a 12-h time step, we used output at an annual time step to fit the VAR model. We restricted the analysis to the most recent 25 years of biomass to allow the model to reach quasi-equilibrium. We also restricted the time series to 37 (of 62) functional groups. Initial exploration revealed that the VAR models struggled to converge for functional groups with drastically different generation times (e.g., whales and zooplankton). Therefore, we focused on lower trophic level functional groups, in this case, prey and prey of prey of the mackerel functional group (including Pacific mackerel Scomber japonicus and jack mackerel Trachurus symmetricus). These modeled biomass time series were then used as responses in a VAR model of the California Current ecosystem (as observation error is not included as part of the Atlantis ecosystem model, we did not apply VARSS models to these data). Combined, these cutoffs yielded 925 data points; fitting this kind of data in a VAR framework where all interactions are possible (e.g., none are fixed a priori at 0) includes 1,406 parameters (1,369 interactions in B, 37 variance parameters in Q).

Instead of just focusing on changes in single interspecific interactions, a broader question of interest is whether ecological communities are stable. To illustrate the impact of regularization on inference about community stability, we used the posterior estimates of B to calculate two metrics proposed by (Ives et al., 2003). First we calculated the proportion of stationary variance attributed to species interactions, det|B|2/m, where m is the number of species in the community. Values of this stability metric greater than 1 indicate unstable systems, and smaller values closer to 0 represent greater stability. Second, we calculated the rate of return, as the dominant eigenvalue of B. We calculated each of the stability metrics separately for each MCMC draw to produce a posterior distribution of stability for each alternative prior formulation.

Results

Simulated data

Our comparison of priors (Normal, Laplace, Regularized horseshoe) to simulated data indicated that posterior distributions of B matrix parameters were qualitatively similar between the Laplace and horseshoe priors, with the latter assigning slightly more density near 0 (Fig. S1). After removing less than 3% of iterations where models had difficulty converging (R-hat > 1.1), the total log-score across all parameters in the B matrix indicated that the horseshoe prior generated estimates that were most accurate and precise (Fig. 2). Some modeling applications may be more or less concerned with estimates of density dependence (diagonal of B), or estimates of species interactions (off-diagonal elements of B). Because of wider tails, the normal and Laplace priors were better able to capture non-zero off-diagonal elements, but were worse at estimating elements of B that were assigned values of 0 (Fig. 2). When standardized to a common scale, the regularized priors do a better job at estimating non-zero elements than the normal prior does at estimating true zeros (Fig. 2). Our LOOIC comparison to quantify the impact of alternative priors on predictive accuracy showed that the Laplace and horseshoe priors were slightly better than the normal distribution (though these estimates have considerable uncertainty; Fig. S2).

Figure 2 Distribution of the average difference in log-scores between alternative priors on B matrix elements and the best model (each box represents 200 replicated datasets, and with values of 0 representing the best model).

Values are averaged to allow comparison between (1) all B parameters, (2) diagonal elements, (3) off-diagonal elements that are not zero, and (4) off-diagonal elements that are zero.

Our simulations used a fixed process variance, and varied the level of observation error variance to explore how signal to noise ratios impact estimates of the B matrix elements. Varying the observation error highlighted that reducing observation error minimizes the differences between priors (or in contrast, increasing observation error makes the least accurate or precise priors even worse).

Application to marine food webs

In our application to data from the California Current marine food web of 37 marine species, there were substantial differences in LOOIC between models with alternative priors; the model with normal priors had the highest estimate (LOOIC = 389.3 ± 45.6), followed by the Laplace prior (222.2 ± 69.4) and finally the model with regularized horseshoe priors (−14.2 ± 65.7). The maximum value of R-hat across parameters and models was 1.024, with the exception of the model with Laplace priors (two parameters between 1.11–1.13). These results indicate that the model with regularized horseshoe priors (lowest LOOIC) has the best approximated out of sample predictive ability.

The effects of regularized B matrix priors are easily seen when comparing estimates from a VAR model with normal priors to one with regularized horseshoe priors (Fig. 3). With regularization, the majority of off-diagonal B matrix elements are assigned values close to 0. Despite zeroing out the majority of species interactions, the VAR model with regularized horseshoe priors appears to identify ecologically important interactions. The 37 components of the food web in our analysis were centered around Pacific mackerel—the interspecific effect that was found to have the largest estimated effect on mackerel biomass is a positive effect of copepods (Fig. 4), an important diet item for this species (Dufault, Marshall & Kaplan, 2009; Brodeur et al., 2019).

Figure 3 Posterior means of species interaction estimates from the B matrix for the California Current marine food web.

Results from using two priors for the off-diagonal elements are shown: a normal prior with each element estimated as a unique parameter, and a regularized horseshoe prior. Diagonal elements generally have a different range (0 – 1) compared to off-diagonal elements.

Figure 4 Time series of standardized biomass for mackerel and copepods, used in the VAR model of the California Current marine community.

The two strongest effects on mackerel biomass are also shown (positive effect of copepods on mackerel, and a slight degree of density dependence of mackerel).

Finally, we compared the posterior distributions of estimated stability across alternative prior formulations. These results indicated that there were slight increases in estimated rates of return moving from horseshoe to Laplace to normal priors (Fig. 5). Rates of return within the unit circle are expected in stationary systems (Ives et al., 2003), and the model with the regularized horseshoe prior appears closest to this assumption. Stability, calculated as det|B|2/37 was more similar across alternative prior formulations, with wide and overlapping credible intervals—though the point estimate for the model with horseshoe priors appeared slightly higher, translating to less stability (Fig. 5).

Figure 5 Posterior distributions of community stability from the VAR models of the California Current marine food web, derived as det(B)2/37 and the dominant eigenvalue of B.

Estimates are shown across a range of potential priors for the off-diagonal elements of B. Boxes represent the posterior quartiles (and median) and the vertical lines represent the upper and lower extremes.

Discussion

As ecological datasets have grown larger and larger, statistical variable selection techniques have also evolved to reduce model complexity and help to identify important covariates. Penalized regression techniques offer several advantages over methods that are currently widely used in ecology (e.g., stepwise and all subsets regression). Regularizing priors, such as the Laplace and horseshoe used here offer several advantages over traditional methods. First, by including hyperparameters, uncertainty in the degree of regularization is propagated into the coefficient estimates. Second, model complexity is implicitly accounted for by setting many of the model coefficients near 0. Unlike stepwise variable selection, which may become trapped in valleys and need to be initialized from multiple starting points, a third advantage is that in the Bayesian approach, a model only needs to be run once (provided MCMC chains indicate convergence). These approaches should be used with caution however, as coefficients in some models may be over-penalized (resulting in many estimates near zero).

Results from our simulated datasets using regularizing priors and VAR models illustrated that because the Laplace or horseshoe priors will result in many posterior estimates near zero, models with those priors are better able to identify true zeros. As expected, advantages of regularizing priors generally diminish as observation error is increased and the signal to noise ratio is decreased (Fig. 2). Depending on whether these small interactions are a focus of inference, or whether the goal of an analysis is to find the VAR or VARSS model with the best predictive ability, the Laplace or regularizing horseshoe may each offer advantages. While we used log-scores to quantify the accuracy and precision of alternative models, other studies may be interested in other types of predictive performance (e.g., out of sample forecasting) and results would be expected to differ slightly depending on the type of inference. Regardless of the application, we recommend analysts compare the results from several formulations of priors, after establishing the goals of the modeling.

Our estimation of interactions among 37 species in the California Current ecosystem represents a more realistic ecological analysis, where the potential complexity of the model exceeds the number of data points. With regularizing priors, posterior estimates of interspecific interactions from the Bayesian VAR model were generally shrunk toward zero (representing weak interactions, Fig. 3). As this food web was constructed with mackerel as a central focus, it is promising that the strongest interspecific effects on mackerel is a positive effect of copepods. Large zooplankton (euphauisiids) are the most important diet item for mackerel in the California Current Atlantis model, however the strength of the copepod result may be driven by both direct and indirect interactions (copepods are a diet item for mackerel, but also are the primary prey of euphauisiids). A similar strong linkage between mackerel and copepods was also found in Kaplan et al. (2013)—they simulated the effects of various levels of fishing pressure on forage fishes and found that scenarios with high exploitation rates of mackerel had a positive effect on euphauisiids, and subsequent negative interactions on copepods. The relationship between mackerel and copepods is also interesting because mackerel represent a generalist predator in the California Current Atlantis model (also consuming juvenile Pacific hake, Merluccius productus, cephalopods, and other species). These smaller interactions were estimated to be much smaller than the effect of copepods however, suggesting that some ecologically important interactions may be estimated as close to zero with this approach. When applied to other large datasets in the future, it may be useful to use different priors for elements of the B matrix thought to be important (as opposed to our approach, which assigned the same prior to all off-diagonal elements).

Ecological applications of multi-species models are increasingly common (Hampton et al., 2013a). For example, they have been used to examine food web dynamics in plankton communities (Ives et al., 2003; Hampton, Scheuerell & Schindler, 2006), analyze effects of shifting climate on large ecosystems (Hampton et al., 2008; Francis et al., 2014), illustrate portfolio effects in coral fishes (Thibaut, Connolly & Sweatman, 2012), and evaluate varying effects of commercial fisheries (Dalton, 2001; Lindegren et al., 2009). Combining regularizing priors with VAR or VARSS time series models offers one approach to simplifying the complexity of a large food web into a smaller number of interpretable components and indicators of emergent properties like stability. Future advances with these models could experiment with the inclusion of sample replicates, known observation errors (via other surveys for example), and time-varying interactions.

Supplemental Information

Supplemental Information 1 Posterior estimates for one replicate of the simulated dataset.

The true parameter value is shown with a vertical red line in each box. Each element corresponds to the species in Table S1, and the interpretation of elements are the effects of species in each column on the species in each row (e.g., Bmat[1,2] is the effect of small phytoplankton on large phytoplankton).

Click here for additional data file.

Supplemental Information 2 Difference in approximate leave one out information criterion (LOOIC) point estimates between the best model vs. alternatives.

For each observation error level, 200 random datasets are compared (value of 0 corresponds to the model with highest predictive ability).

Click here for additional data file.

Supplemental Information 3 Interaction matrix from Ives et al. (2003) used for simulations.

Click here for additional data file.

The authors would like to thank M. Betancourt and J. Arnold for developing clear vignettes and case studies with guidelines around Bayesian regularization. KM was supported on a National Research Council (NRC) post-doctoral fellowship at the Northwest Fisheries Science Center while this research was performed. We thank I. Kaplan for providing helpful edits and discussions about the Atlantis model, as well as to B. Fulton for Atlantis model development. Any use of trade, firm, or product names is for descriptive purposes only and does not imply endorsement by the U.S. Government.

Additional Information and Declarations

Competing Interests

Author Contributions

Data Availability

Eric J. Ward is an Academic Editor for PeerJ.

Eric J. Ward conceived and designed the experiments, performed the experiments, analyzed the data, prepared figures and/or tables, authored or reviewed drafts of the article, and approved the final draft.

Kristin Marshall conceived and designed the experiments, analyzed the data, prepared figures and/or tables, authored or reviewed drafts of the article, and approved the final draft.

Mark D. Scheuerell conceived and designed the experiments, analyzed the data, prepared figures and/or tables, authored or reviewed drafts of the article, and approved the final draft.

The following information was supplied regarding data availability:

The code is available at GitHub: https://github.com/atsa-es/varlasso; Eric Ward. (2022). atsa-es/varlasso: PeerJ submit (v1.1.1). Zenodo. https://doi.org/10.5281/zenodo.6857994.

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
