# Peer review of "Regularizing priors for Bayesian VAR applications to large ecological datasets"

_PeerJ, doi:10.7717/peerj.14332_

## Round 0.1 · original submission · Minor Revisions

This is a valuable contribution to community ecology and the world of modeling large, complex datasets. The 2 reviewers have made useful suggestions. Please address each of the reviewer comments in your resubmission.

·

Basic reporting

The article is clear, well structured, and well written. The background is well described and supported with cited literature. I have no suggested improvements, other than some minor typos:
Line 48 “priors to a output”
Line 136 “alternatives continuous priors”
Line 164 “to also incorporate and observation”

Experimental design

The research gap is well defined, and the priors are robustly tested and compared. The method is well described and model code is supplied in an R package. I have no suggested improvements.

Validity of the findings

The findings are clear and well reported. Commented-out code in the supplied R package in the 'fit' script provides example data to fit the VAR-lasso model.

The following are minor comments - and I don't consider revision essential. One area where I wanted more was additional insight into the importance of regularized priors for typical ecological analyses using VAR - such as their example food webs. The value of regularization seems in large part to boil down to whether I want to identify most including weak interactions? Does shrinking these better identify important interactions (Fig. 3 suggests yes)? Confidence in the remaining interactions after penalization might be increased by the authors focusing on whether regularization missed some key expected interactions, and not just on the expected interaction it did identify (copepods).

Relatedly: (line 356) “Penalized regression techniques offer several advantages over methods that are currently widely used in ecology (e.g., stepwise and all subsets regression). Regularizing priors, such as the Laplace and horseshoe used here offer several advantages over traditional methods.” How would these regular methods translate to the food web analysis? Are they even a realistic option for food web analyses? I don't see why someone would choose to exclude species they thought weren't important before running a VAR model, or identify a valid hierarchy to test. Instead, is regularization the only realistic option for 'dimension reduction' for these types of ecological analyses? I’m thinking about these methods for normal regression, and perhaps I don’t appreciate how this works for VAR.

Additional comments

Some minor additional comments that could improve clarity:

Line 282: “We restricted the analysis to the most recent 25 years”. Some clarification could be useful here. Does most recent to current mean that the 50 years were grounded in reality (e.g. actual or projected catches)? So are the "most recent" the first 25 years or the latter 25 years of the series?

“Bias” is mentioned in the abstract and methods, but not in results when reporting on Fig. 2.

Line 331, suggest changing Gaussian to normal for consistency.

The narrow distributions of the horseshoe prior can be difficult to see in Fig. S1, especially given the grey grid lines.

·

Basic reporting

Basic reporting is mostly clear and concise.
The discussion is missing some of the results (e.g. effect of observation and process error, LOOIC results, ecosystem stability).
Ecosystem stability was not mentioned in the introduction, so when I came across it in the methods it felt ‘tacked on’. I think it is fine to include it, but just bring it into the introduction, and also the discussion.

Experimental design

The experimental design seems to be sound as described in the methods, except perhaps for the assumed stationarity. I would prefer a test applied to ensure stationarity rather than this being assumed based on what was deemed to be a sufficiently long burn-in period.

Validity of the findings

The findings seem valid, although as the MCMC convergence diagnostics have not been presented, it is difficult to tell.

Additional comments

Generally I thought this was a nice piece of work. My comments mostly relate to expanding further on what has been written in some areas. A few of the sections in the results and methods did not make it into the discussion. MCMC convergence diagnostics were not shown, and it is important that these are included, as poor convergence would undermine any results presented.

Please see included annotated pdf for full detailed suggested edits.

---

## Round 0.2 · accepted · Accept

Thank you for addressing all of the reviewer's comments and providing an updated manuscript. I reviewed the manuscript and find it acceptable, but there a few very minor typographical errors that should be fixed at the proof stage.

Extra space in the abstract, line 39

Two incomplete references:
Barton K. 2020. MuMIn: Multi-Model Inference.
Miller TL based on F code by A. 2020. leaps: Regression Subset Selection.